# Cancer in Migrants: A Population-Based Study in Italy

**DOI:** 10.3390/cancers15123103

**Published:** 2023-06-08

**Authors:** Giulia Collatuzzo, Margherita Ferrante, Antonella Ippolito, Alessia Di Prima, Cristina Colarossi, Salvatore Scarpulla, Paolo Boffetta, Salvatore Sciacca

**Affiliations:** 1Department of Medical and Surgical Sciences, University of Bologna, 40138 Bologna, Italy; giulia.collatuzzo@studio.unibo.it; 2Department of Medical, Surgical and Advanced Technologies “G.F. Ingrassia”, University of Catania, 95123 Catania, Italy; 3Integrated Cancer Registry of Catania-Messina-Siracusa-Enna, University of Catania, 95123 Catania, Italy; 4Mediterranean Institute of Oncology (IOM), 95029 Catania, Italy; 5Stony Brook Cancer Center, Stony Brook University, Stony Brook, NY 11794, USA; 6Department of Family, Population and Preventive Medicine, Renaissance School of Medicine, Stony Brook University, Stony Brook, NY 11794, USA

**Keywords:** migrants, cancer epidemiology, Italy

## Abstract

**Simple Summary:**

This study investigates cancer in migrants in Southern Italy, who represent a neglected but vulnerable population. We used data from the Eastern Sicily Cancer Registry collected between 2004 and 2019 to compare the adjusted proportionate morbidity ratio for the most common cancer types in migrants and non-migrants, and we calculated the odds of migrant status for one cancer compared to all cancers. The migrants/non-migrants odds of cancer was 2.1%, with most cancers occurring in migrant women. We observed increased proportions in cervical and lung cancer, with higher odds of cervical cancer and lower odds of colorectal cancer in migrants. Measures should be implemented to enhance the access of migrants to prevention, early diagnosis and care for cancer. These interventions should account for the migrant’s country of origin. Particular attention should be given to HPV vaccination, cervical cancer screening and tobacco control to reduce the cancer burden in this population.

**Abstract:**

Background: Migrants are a vulnerable and neglected population. We aimed at investigating cancer proportionate rates in migrants in Sicily, Southern Italy. Methods: We extracted data on new cancer cases diagnosed between 2004 and 2019 from the Eastern Sicily cancer registry. We compared the adjusted proportionate morbidity ratio (PMR) for the most common cancer types among migrants and non-migrants. We fitted multivariate logistic regression models comparing one cancer to all other cancers to calculate odds ratios (ORs) and 95% confidence intervals (CIs) for migration status. The analysis was stratified by region of origin. Results: Overall, 4726 new cancer cases occurred in migrants between 2004 and 2019, 63.5% of those among women and 224,211 in non-migrants, including 54.5% among men, with odds for migrants/non-migrants of 2.1%. Migrants had an increased proportion of cervical (PMR = 2.68, 95% CI = 2.29–3.10) and lung cancer (PMR = 1.20, 95% CI = 1.07–1.33). The highest OR in migrants was observed for cervical cancer (OR = 3.54, 95% CI = 2.99–4.20). Colorectal cancer was decreased among migrants (OR = 0.86, 95% CI = 0.77–0.96). Conclusions: Migrants to Sicily have higher odds of cervical cancer and a decreased risk of colorectal cancer compared to non-migrants. Increased odds were also detected for lung cancer, in particular in women. Different cancer patterns could be observed based on the region of origin. HPV-related cancers need targeted attention in migrants living in Sicily.

## 1. Introduction

Migrants represent a vulnerable subgroup of the population from multiple points of view [1], including socioeconomic condition (job position, income and educational level) [2] as well as health status (vaccination history, participation in screening programs, disease diagnosis and care). Migrants often suffer disease disparities [3,4], such as poor disease management, lack of check-ups and screening tests [5], delayed diagnosis [6] and inadequate treatments. A meta-analysis reported that foreign-born women were less likely to be diagnosed with localized-stage breast cancer compared to native women [6], including a relationship between the magnitude of the disparity and the level of development of the country of origin. On the other hand, reports have been published in which migrants registered a lower risk of cancer and lower cancer mortality compared to non-migrants [7,8,9], possibly due to a healthy migrant effect and low incidence of some cancers in the country of origin.

The number and proportion of cancer deaths and incident cases in migrants have been estimated in several countries, providing quantitative evidence of cancer epidemiology in this population group [8,9,10,11]. Cancer epidemiology in migrants is important for several reasons [3,4,6]. First, the description of cancer in this special population offers valuable information to understand the determinants of cancer in different ethnic groups and to disentangle the role played by environmental and genetic factors. Additionally, the investigation of cancer in migrants can highlight patterns of cancer occurrence in different ethnic groups, possibly identifying subjects to be targeted for specific preventive actions [3,12,13]. Moreover, studies on cancer in migrants may help in understanding the causes of cancer also in native populations [12]. These results may also have implications for national regulations and health policies, providing useful information derived from the comparison between the epidemiologic data of cancer in people born in different countries [14]. Indeed, migrants may acquire the same risk profile of the population of the host country [15]. An example comes from a population-based study conducted in Norway, which reported higher overall cancer incidence rates in native people than in migrants and observed higher liver cancer incidence in Asians than in Norwegians, as well as higher lung cancer incidence in male migrants from other Nordic countries and from Eastern Europe than in native men [9]. 

The Mediterranean countries of Europe are subject to migration from Northern Africa, Eastern Europe and West Asia, given the geographical position [14]. A recent study reported declining trends in cancer mortality in migrants in Spain between 2000 and 2016 [14]. In general, however, data on cancer incidence rates in migrants to Mediterranean countries in the last decade are scarce. 

Sicily is an island in Southern Italy, which has experienced an increase in the migrant population in recent years. The official proportion of foreign subjects over the total population in Sicily in 2021 was 3.9%. The main countries of origin were Romania. Tunisia, Morocco, Sri Lanka, Albania and Bangladesh [16]. The actual number of migrants, however, is likely to be higher because of the presence of illegal and seasonal migrants. This is particularly true for Sicily, because of (i) the widespread use of seasonal migrants in agriculture and (ii) the role of Sicily as an entry point from the Mediterranean and the presence of numerous temporary transit camps for undocumented and illegal immigrants.

In order to estimate cancer proportion in migrants at a population-base level, we analyzed data of a cancer registry in Sicily, Italy. We focused on major areas of origin of the migrants and on the cancer sites with the highest occurrence in this special population. 

## 2. Materials and Methods

### 2.1. Study Design and Population

This study is designed as a case–control study, where migrant status is the exposure and cancer types are the controls. 

We analyzed data from the Eastern Sicily Cancer Registry covering 2.5 million people from four provinces (Catania, Enna, Messina and Syracuse) [17]. The registry is considered to be complete and is included in the Cancer Incidence in Five Continents, a collection of high-quality registries maintained by the International Agency for Research on Cancer [18]. This registry includes cases identified only from death certificates. Data are validated and periodically checked by the Associazione Italiana Registro Tumori (AIRTUM) through different programs (e.g., CheckAIRTUM and IARC CRG Tools). We selected new cases of the most frequent cancers diagnosed between 2004 and 2019 in migrants and identified new cases of the same cancers occurring among non-migrants during the same time period. The Cancer Registry collects data on cancer diagnosed mostly based on histological confirmation of the primary tumor and, for a minority of cases, clinical based on data or imaging; during 2009–2012, for only 1.6% of new cancer cases in men and 2.1% in women, the site of origin was classified as ‘other or unspecified’ [18]. 

### 2.2. Data Sources

Data derived from the Eastern Sicily Cancer Registry. We extracted the following information from the Cancer Registry: sex, age, country of birth, basis for diagnosis (histology/citology; clinical; death certificate/other), date of diagnosis and treatment (chemotherapy, radiotherapy and surgery). Information on the three treatment modalities was missing for a proportion of subjects. Since we were not able to distinguish between missing information or no therapy, we did not include these data in the analysis. Country of birth was categorized as Northern/Western Europe, Eastern Europe and Balkans, Northern Africa, Sub-Saharan Africa, Western Asia, other Asian countries excluding Japan, North America/Oceania/Japan and Latin America. 

### 2.3. Statistical Analysis

Information on the total number of migrants living in the four provinces covered by the Cancer Registry was not available. We, therefore, could not calculate incidence rates and ratios directly comparing migrants and non-migrants but, rather, used the proportions of cancer occurring in the two populations. Specifically, we calculated the proportion of new cases of each cancer over total cancers among migrants and compared this with the same proportion among non-migrants. We then calculated the proportionate morbidity ratio (PMR) for each cancer type as the ratio of new observed cases in migrants over new expected cases, based on the proportion in non-migrants after adjusting for sex, age group and calendar year. Further, 95% confidence intervals (CIs) of PMR were calculated based on the Poisson distribution of new expected cases. We stratified the analyses by region of origin, sex and age. We tested heterogeneity in PIR between geographic region, sex and age categories using the Cochrane Q-test [19].

In addition, we fitted multivariate logistic regression models comparing one cancer to all other cancers to calculate ORs and 95% CIs of migration status (overall and by sex), after adjustment for sex, age category, basis of diagnosis and period of diagnosis. We repeated the analysis by geographic region of origin by fitting separate models, including migrants from one specific region and all non-migrants, and after stratification, by period of diagnosis. We tested heterogeneity between strata of sex and age period by adding interaction terms to the regression models.

For all the aforementioned analyses, *p* <0.05 was considered statistically significant. 

## 3. Results

Table 1 illustrates the main characteristics of the study population. Overall, a total of 4726 new cases of cancer were registered among migrants between 2004 and 2019, including 1724 (36.5%) new cases among men and 3002 (63.5%) new cases among women. In the same period, 224,211 new cases were registered among non-migrants, including 122,241 (54.5%) among men and 101,970 (45.5%) among women. The overall odds of new cases in migrants to non-migrants was 2.1% and increased from 1.7% in 2004–2007 to 2.5% in 2016–2019. The countries of origin with the largest number of new cases of cancer among migrants were Germany (N = 968), Libya (N = 625) and Romania (N = 442).

Overall and sex-specific PMR for the main cancer types is shown in Table 2. We observed an increased proportion of cervical cancer (PIR = 2.68, 95% CI = 2.29–3.10) and lung cancer (PIR = 1.20, 95% CI = 1.07–1.33) among migrants. The result of lung cancer for both sexes (PMR = 1.20, 95% CI = 1.07–1.33) was driven by the pattern in women (PMR = 1.32, 95% CI = 1.11–1.56), with no increased proportion in men (*p*-value of test of heterogeneity between sexes = 0.09). The PMR of leukemia was decreased, with a stronger result among women (PMR = 0.77, 95% CI = 0.61–0.95, *p*-heterogeneity between sexes = 0.07).

The results of the multivariate logistic regression analysis (Table 3) are consistent with the previous ones. The cancer with the highest OR in migrants was cervical cancer (OR = 3.54, 95% CI = 2.99–4.20), and an increase was also detected for lung cancer, in particular in women (OR = 1.23, 95% CI = 1.03–1.47). Colorectal cancer was the only neoplasm whose OR was decreased among migrants (OR = 0.86, 95% CI = 0.77–0.96), and a decreased OR of borderline statistical significance was observed for liver, breast and prostate cancer and NHL. Liver cancer and leukemia were the two neoplasms for which there was evidence of heterogeneity in OR between men and women (*p* = 0.02 and 0.03, respectively).

In the analysis by region of origin in migrants (Figure 1, detailed results are available in Appendix A), migrants from Northern and Western Europe, including the European Union, showed a decreased OR for colorectal and breast cancer as well as NHL and leukemia, and an increased OR for cervical cancer. Migrants from Eastern Europe showed a decreased OR for prostate cancer and an increased OR for lung and cervical cancer. Migrants from North Africa experienced an increased OR for bladder cancer, whereas migrants from Sub-Saharan Africa experienced an increased OR of liver, breast cancer and leukemia and a decreased OR for colorectal cancer. The results on migrants from West Asia were hampered by small numbers. Migrants from other Asian countries had a decreased OR for bladder cancer and an increased OR for leukemia. Migrants from North America, Oceania and Japan did not have a statistically significant increased or decreased OR for any cancer. Finally, the OR for breast cancer was increased among migrants from Latin America.

The analysis by period of diagnosis suggested a trend in OR of liver cancer (OR increased from 0.55 (95% CI 0.30–0.98) in 2004–2007 to 1.13 (95% CI 0.78–1.65) in 2016–2019), lung cancer (OR increased from 0.89 (95% CI 0.67–1.19) to 1.22 (95% CI 1.00–1.49)) and breast cancer (OR increased from 0.84 (95% CI 0.68–1.04) to 1.08 (95% CI 0.94–1.25)), although none of these trends were statistically significant (results not shown in detail).

## 4. Discussion

Our analysis revealed several patterns of cancer incidence among migrants in a Southern Italy population, including a higher proportion of cervical and lung cancers and a borderline statistically significant lower proportion of breast and prostate cancers compared to non-migrants. The stratification by geographical region of origin revealed that these patterns were mainly due to migration from Europe.

The results we describe are impaired by the lack of population data on the number of migrants in Sicily, thus preventing us from calculating cancer incidence rates in this population and incidence ratios in the comparisons with non-migrants. We tried to address this problem by using official data on the number of migrants present in four provinces during the study period [20] but obtained unreliable results, likely due to an undercount of migrants in official statistics. Despite this important limitation, ours remains one of the few studies to provide data on the neglected issue of cancer incidence in migrants in Italy and one of the first to provide a comprehensive analysis of different cancer types in migrants.

Although there was no increased proportion of head and neck cancer among migrants, an increase was suggested for migrants from Eastern Europe, the Balkans and from Asia, and a decrease was found among migrants from sub-Saharan Africa and Latin America. Possible explanations are relatable to the different distribution of the risk factors of head and neck cancer in Italy-born people and migrants, specifically HPV, tobacco smoking and alcohol [21]. Central and Eastern Europe is one of the regions with the highest incidence of this group of cancers [22].

Incidence of gastric cancer is elevated in Eastern European and East Asian countries [23], and migrants from these regions had a higher proportion of gastric cancer, although the difference was not statistically significant. Conversely, the incidence of colorectal cancer is relatively low in sub-Saharan Africa [24], and migrants from these countries had a non-significantly lower proportion of colorectal cancer. We observed a reduced risk of liver cancer in migrant men but not in women. Despite not having the information to assess the reason of this sex difference, we may hypothesize that it depends on sociocultural factors, leading to a better management of chronic liver disease (which can be a precursor of cancer) in men than in women, e.g., more frequent clinical visits and medical exams. However, this neoplasm was increased among migrants from Sub-Saharan Africa, a region at high risk for hepatitis B infection and liver cancer [25,26].

We observed an increased proportion of lung cancer among migrants, which was primarily related to migrants from Eastern European countries. The high proportion of lung cancer in migrants could be explained by higher exposure to risk factors, including tobacco smoking, indoor and outdoor air pollution and occupational risk factors during their lifetime. The difference between women and men is likely due to the low incidence of lung cancer among women in Southern Italy [18], which was explained by low tobacco consumption in past decades. Thus, the sex pattern observed in migrants may be an artifact rather than reflect particular risk factors in migrant women, despite the fact that we could not exclude potential confounders.

The proportion of breast cancer was reduced among migrants from Eastern Europe and the Balkans, a region with a lower incidence of these neoplasms compared to Italy [27]. A similar pattern was shown for migrants from other countries of Europe, while migrants from sub-Saharan Africa and Latin America had a higher proportion of this neoplasm. These differences by region of origin may derive from different approaches to cancer screening in addition to different exposure circumstances. In particular, among the factors which may affect breast cancer risk, oral contraceptives have been reported to be higher in women from Eastern Europe. 

Cervical cancer showed the highest difference between migrants and non-migrants of all cancers, which was statistically significant for migrants from Europe. This is not unexpected, given the low incidence of this disease in Southern Italy and Sicily in particular [18]. The fact that the greatest proportion of new cases was seen among migrants from Eastern Europe might be explained by the high prevalence of HPV infection in that region [28]. An Italian study reported that 58% Eastern European and African women vs. 19% of Italy-born women to be HPV-positive [29]. Campari et al. reported a higher prevalence of preneoplastic cervical lesions and a lower participation in cervical cancer screening among migrants than Italian women [30].

The proportion of prostate cancer was lower in migrants from Eastern Europe and the Balkans and, although not significantly so, from Asian countries, excluding West Asia, which is consistent with previous findings [31,32]. This difference may be attributable to different levels of “westernization” of the lifestyle habits in different geographical areas of the Asian continent [33]; however, random fluctuation may also explain the difference we observed. Further, the results on the higher incidence rate of bladder cancer in migrants from North Africa than non-migrants are consistent with worldwide patterns of this disease [34]. 

These heterogeneous patterns of risk of specific cancers in migrants indicate the need for tailored cancer control programs based on the specific cancer predisposition in different populations. Overall, these results agree with a review by Arnold and colleagues, which described a higher proportion of infectious-related cancer (e.g., gastric and cervical) and a lower proportion of lifestyle-related cancer (e.g., colorectum, breast and prostate) in migrants to Europe [35]. 

Our results provide novel evidence on the different incidence of cancer in migrants in Italy and Sicily in particular, and they relate the pattern of cancer occurrence by geographical area [27]. The results obtained reflect lifestyle, environmental and genetic factors, which underlie the occurrence of specific cancers, such as cervical and prostate cancer in the Balkans [36] and colorectal cancer in Africa [24,37]. Compared to non-migrants, migrants from Eastern Europe and Africa may be less sedentary [38] and have a healthier lifestyle [39], including diet [40,41,42], and a lower prevalence of dysmetabolic diseases, such as diabetes and hypercholesterolemia [43], factors which may be associated with a lower incidence of prostate [44] and colorectal cancer [45]. Conversely, non-migrants may be less exposed than migrants to unsafe sex, resulting in a lower prevalence of HPV infection and HPV-related cancers [46,47,48], a pattern which has also been observed in other countries [49,50].

Migrants represent, in large part, a vulnerable group, connoted by a higher prevalence of unhealthy lifestyle habits (e.g., tobacco smoking, alcohol drinking [22], poor diet [51]), occupational disparities [2] and reduced access to healthcare services (e.g., vaccination, screening) [52,53]. These aspects are ultimately related to socioeconomic status. Most of the migrant populations, in Italy as well as in other countries, belong to low socioeconomic status. The relationship between low socioeconomic status and cancer is well described [54], especially with regard to some types of cancers, including those for which we evidenced an increased PIR, namely cervical [55,56] and lung [57,58,59]. 

Our analysis addresses the issue of cancer in migrants in Sicily through two approaches, namely PIR estimates and multivariate logistic regression analysis. This latter could account for several aspects, including treatment, which has been reported to differ between migrant and non-migrant populations. Different cancer treatment causes cancer disparities in migrants and non-migrants and is linked to barriers experienced by migrants, such as lack of language proficiency and not being familiar with the health system [60]. Our data on treatment, however, were impaired by missing data. This limitation is unlikely to have introduced bias, because even if missing data could have led to misclassification of treatment, we observed no confounding effect by this variable in the multivariate model. 

When focusing on results by geographical area, an important observation is the increased proportion of breast cancer in women from sub-Saharan Africa. This result is unexpected, as African populations (incidence rates around 50/100,000) usually have lower incidence rates of breast cancer compared to European populations (incidence rates 80–90/100,000) [61]. The reason for this result is not clear. Breast cancer screening is usually associated with higher numbers of diagnoses [62]; thus, disparities in participation for cancer screening among migrants do not seem to explain our finding [63]. In addition to this, the migrant population may not precisely reflect incidence rates of the country of origin because they do not represent a random sample of that population. Interestingly, the increased proportion of breast cancer in this migrant subgroup is homogeneously distributed by age. 

This study has some limitations. First, as already mentioned, the number of migrants in Sicily was not available, preventing us from calculating the incidence of cancer in migrants and comparing it with that in non-migrants. For this reason, the study design is that of a case–control study where controls are cancers in non-migrants. This approach has been used by other authors [64]. In addition, no information was available on the duration of residence of the migrants: analyses by duration of stay in the host country help to clarify the role of factors affecting cancer risk [65]. Moreover, the lack of information on lifestyle, occupational and sociodemographic factors prevented us from adjusting and stratifying for important variables. Additionally, results by area of origin of the migrant population showed patterns of risk which were not fully consistent with previous literature. Last, small numbers impaired the stratified analyses for some areas of origin.

The present study also has several strengths. We provided updated data on an under-investigated topic, focusing on a special population which is particularly vulnerable and affected by health inequalities. We identified interesting cancer patterns, and, because of the population-based nature of our data, we added valuable data to the current literature on cancer epidemiology in migrants and offered new important information on cancer incidence in different groups of migrants in Italy compared to non-migrants. Further, the data we used were from a high-quality cancer registry, increasing the reliability of our results [66,67]. 

## 5. Conclusions

In conclusion, migrants to Sicily appear to have an increased OR of cervical and lung cancer than non-migrants, although the comparison is based on proportionate ratios. Different cancer patterns could be observed based on the area of origin of the migrants, with lifestyle and socioeconomic factors in migrants from Centra/Eastern Europe, the Balkans and Africa possibly explaining several results. Differences were identified in particular for women, regarding HPV-related cancers as well as lung cancer and hematologic malignancies. These data may be a useful source of information for understanding cancer epidemiology in migrants in Italy. Cancer control in this special population requires public attention, despite its future trends being unpredictable given the acute nature of migration in Sicily, where most migrants move quickly to other countries.

Given the vulnerability of migrants to cancer, and the discrimination which they might be subjected to, measures should be implemented to enhance their access to prevention, diagnosis and care for cancer. Tailored intervention may be developed based on migrants’ country of origin, given the different cancer risk by geographical area. Targeted attention to HPV vaccination and cervical cancer screening participation in migrant women would help to better control infection-related cancers. Tobacco control interventions targeting migrants would also be important to reduce the cancer burden in this population. Cancer control and early detection in migrants may improve, while it is difficult to establish a follow-up for such a special population, which is quickly moving from Sicily to other countries. 

## Figures and Tables

**Figure 1 cancers-15-03103-f001:**
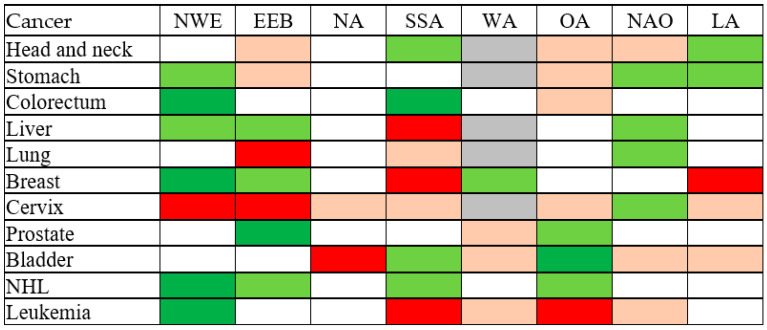
Odds ratio of selected cancer among migrants, by area of origin—results of multivariate logistic regression analysis. NWE, Northern and Western Europe and European Union, excluding Bulgaria and Romania. EEB, Eastern Europe and the Balkans. NA, North Africa. SSA, Sub-Saharan Africa. WA, West Asia. OA, Other Asia, excluding Japan. NAO, North America and Oceania, including Japan. LA, Latin America. Light green: OR < 0.8, *p* > 0.05. Dark green: OR < 1, *p* < 0.05. Light red: OR > 1.25, *p* > 0.05. Dark red: OR > 1, *p* < 0.05. Grey: Model did not converge. NHL, non-Hodgkin lymphoma. OR, odds ratio adjusted for sex, age, type of diagnosis, year of diagnosis.

**Table 1 cancers-15-03103-t001:** Distribution of new cases of cancer by migrant status and selected characteristics.

Characteristics	Migrants (%)	Non-Migrants (%)
Total	4726 (100.0)	224,211 (100.0)
Sex		
Men	1724 (36.5)	122,241 (54.5)
Women	3002 (63.5)	101,970 (45.5)
Age group (yrs)		
<55	2165 (45.8)	43,294 (19.3)
55–64	947 (20.0)	42,317 (18.9)
65–74	807 (17.1)	62,606 (27.9)
>=75	807 (17.1)	75,993 (33.9)
Year of diagnosis		
2004–2007	880 (23.2)	52,274 (23.3)
2008–2011	1037 (21.4)	55,360 (24.7)
2012–2015	1330 (28.1)	57,930 (25.8)
2016–2019	1479 (31.3)	58,647 (28.2)
Base of diagnosis		
Histology/cytology	4183 (88.5)	195,654 (87.3)
Clinical	482 (10.2)	25,230 (11.3)
Other ‡	61 (1.3)	3327 (1.5)
Region of origin		
Northern and Western Europe *	1931 (41.1)	-
Eastern Europe, Balkans	751 (16.0)	-
North Africa	890 (18.9)	-
Sub-Saharan Africa	216 (4.6)	-
West Asia	25 (0.5)	-
Other Asian countries †	207 (4.4)	-
North America, Oceania	312 (6.6)	-
Latin America	372 (7.9)	

‡ Including death certificate only. * Including European Union except Bulgaria, Romania (included in Eastern Europe and Balkans). † Excluding Japan (included in North America, Oceania).

**Table 2 cancers-15-03103-t002:** Proportionate morbidity ratios of cancer among migrants by sex.

	Total			Men			Women		
Cancer	N	PMR	95% CI	N	PMR	95% CI	N	PMR	95% CI
Head and neck	81	1.11	0.88–1.37	55	1.11	0.84–1.43	26	1.11	0.74–1.60
Stomach	94	1.00	0.81–1.21	46	1.02	0.75–1.34	48	0.98	0.72–1.28
Colorectum	361	0.92	0.82–1.01	144	0.90	0.76–1.05	217	0.93	0.81–1.05
Liver	81	1.09	0.87–1.35	44	0.99	0.73–1.32	37	1.24	0.89–1.69
Lung	331	1.20	1.07–1.33	194	1.12	0.97–1.28	137	1.32	1.11–1.56
Breast	790	0.94	0.87–1.01	-	-	-	790	0.94	0.87–1.01
Cervix	165	2.68	2.29–3.10	-	-	-	165	2.68	2.29–3.10
Prostate	165	0.94	0.80–1.09	165	0.94	0.80–1.09	-	-	-
Bladder	238	1.06	0.93–1.20	173	1.06	0.91–1.23	65	1.06	0.82–1.34
NHL	182	0.91	0.78–1.05	87	0.98	0.78–1.20	95	0.86	0.69–1.04
Leukemia	155	0.87	0.73–1.01	76	0.99	0.78–1.22	79	0.77	0.61–0.95

N, observed number of new cases among migrants; PMR, proportionate morbidity ratio; CI, confidence interval; NHL, non-Hodgkin lymphoma.

**Table 3 cancers-15-03103-t003:** Odds ratio of selected cancer for migrant status, overall and by gender—results of multivariate logistic regression analysis.

	Total Population	Men	Women
Cancer	OR (95% CI)	OR (95% CI)	OR (95% CI)
Head and neck	1.09 (0.87–1.35)	1.14 (0.88–1.48)	1.05 (0.71–1.45)
Stomach	0.89 (0.73–1.09)	1.01 (0.76–1.35)	0.82 (0.62–1.09)
Colorectum	0.86 (0.77–0.96)	0.86 (0.73–1.03)	0.89 (0.77–1.02)
Liver	0.80 (0.64–1.01)	0.68 (0.49–0.94)	1.11 (0.80–1.55)
Lung	1.12 (1.00–1.26)	1.03 (0.88–1.20)	1.23 (1.03–1.47)
Breast	0.95 (0.88–1.04)		0.95 (0.88–1.04)
Cervix	3.54 (2.99–4.20)		3.54 (2.99–4.20)
Prostate	0.87 (0.74–1.03)	0.87 (0.74–1.03)	
Bladder	1.07 (0.94–1.23)	1.10 (0.94–1.29)	1.05 (0.82–1.35)
NHL	0.86 (0.73–1.01)	0.90 (0.71–1.16)	0.87 (0.69–1.05)
Leukemia	1.02 (0.87–1.20)	1.26 (0.99–1.59)	0.90 (0.71–1.13)

OR, odds ratio adjusted for sex, age, type of diagnosis, year of diagnosis. CI, confidence interval. NHL, non-Hodgkin lymphoma.

## Data Availability

Data can be provided to external investigators upon reasonable request and agreement of the institutions involved.

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
