# Peer review of "Cancer in Migrants: A Population-Based Study in Italy"

_cancers, 2023, doi:10.3390/cancers15123103_

Round 1

Reviewer 1 Report

The authors report a descriptive study in which they analyzed the data on cancer cases among immigrants in comparison to those of local people from 4 provinces in Sicily, Italy.  Two summary statistics were used in the analysis.  One is the ratio of a proportion of a specific cancer of interest among all cancers in migrants against the similar proportion in non-migrants. Another is the odds ratio (OR) of one cancer against all other cancers between migrants and non-migrants. The OR was calculated with multivariate logistic regression in which important covariates and confounders were adjusted. These calculations are fundamentally different from those of conventional ones where the denominators are populations at risk with and without the exposure of interest, and the populations in the conventional calculation are relatively stable over time in terms of their total numbers and demographic compositions.  However, these conditions can be not held here for migrant population. Because of the unstable migrant population and inaccessible statistics and demographics from this population, the authors used ratios to overcome the challenge. Given this change, one can only assume that these summary statistics reflect the relative weight or contribution of one cancer site against all other cancers between migrants and non-migrants. These statistical parameters do not equal to the estimate of risk or incidence that we normally interpretate in epidemiological studies.

If the comments are acceptable to the authors, then the distinction of the ORs and PIRs in this study as opposed to those in the conventional epidemiology should be explained and emphasized in the manuscript, and the study results should not be interpreted as cancer risk or incidence for immigrants.

Author Response

The authors report a descriptive study in which they analyzed the data on cancer cases among immigrants in comparison to those of local people from 4 provinces in Sicily, Italy.  Two summary statistics were used in the analysis.  One is the ratio of a proportion of a specific cancer of interest among all cancers in migrants against the similar proportion in non-migrants. Another is the odds ratio (OR) of one cancer against all other cancers between migrants and non-migrants. The OR was calculated with multivariate logistic regression in which important covariates and confounders were adjusted. These calculations are fundamentally different from those of conventional ones where the denominators are populations at risk with and without the exposure of interest, and the populations in the conventional calculation are relatively stable over time in terms of their total numbers and demographic compositions.  However, these conditions can be not held here for migrant population. Because of the unstable migrant population and inaccessible statistics and demographics from this population, the authors used ratios to overcome the challenge. Given this change, one can only assume that these summary statistics reflect the relative weight or contribution of one cancer site against all other cancers between migrants and non-migrants. These statistical parameters do not equal to the estimate of risk or incidence that we normally interpretate in epidemiological studies.

If the comments are acceptable to the authors, then the distinction of the ORs and PIRs in this study as opposed to those in the conventional epidemiology should be explained and emphasized in the manuscript, and the study results should not be interpreted as cancer risk or incidence for immigrants.

Answer: We agree that our measures do not equal estimates of risk. We carefully checked the text and removed any reference to risk or incidence, and rather used “proportionate morbidity ratio (PMR)”. However, we respectfully disagree with the reviewer that our approach is inherently biased. The use of comparing one cancer against all other cancers (used as controls) has been used in previous case-control studies including multiple cancers, see for example Siemiatycki J, Day NE, Fabry J, et al. Discovering carcinogens in the occupational environment: a novel epidemiologic approach. J Natl Cancer Inst 1981;66:217-25. With this approach odds ratios are underestimated if some of the cancers included in the control pool are also associated with the exposure of interest, and we mention this potential bias in the limitations. Therefore, the odds ratios we calculated were comparable to those used in case-control studies in general.

Reviewer 2 Report

The paper presents relative cancer incidence among migrants to the island of Sicily, Italy.  Using cancer registry data 2004-2019 from Eastern Sicily and examining the most frequently diagnosed cancers, the study found that migrants had increased proportions of cervical and lung cancer but lower proportions of colorectal cancer when compared to the cancer distribution of the local population.

Introduction, first and second paragraphs: touches on the implications of understanding of cancer incidence in this vulnerable group- their direct needs with respect to healthcare/services, as well as potentially missed diagnoses, could be highlighted.

Figure 1, please consider the presentation here- perhaps a forest plot could be more helpful, or consider adding the ORs and CIs to the table cells.  The key is difficult to discern as there seems to be only one shade of red in the table, while it specifies a light and a dark shade of red for increased ORs which are not and are statistically significant (p<0.05).

In the last paragraph of the results- where risks are reported by time period, please add the confidence intervals for all stated ORs.

So that the choice of analytical approach is clear- i.e. why incidence rates were not calculable-, consider adding the statement on lines 210-212 to the methods.

As different patterns are seen by different countries of origin, has there been- or are there anticipated to be- shifts in the origins of migrants to Sicily over the years?  If so, is this worthy of comment that the observed cancer patterns are reflective of the present situation but may change in the future?

Author Response

Introduction, first and second paragraphs: touches on the implications of understanding of cancer incidence in this vulnerable group- their direct needs with respect to healthcare/services, as well as potentially missed diagnoses, could be highlighted.

 Answer: Thank you for the useful suggestion, we added these points to the introduction.

Figure 1, please consider the presentation here- perhaps a forest plot could be more helpful, or consider adding the ORs and CIs to the table cells.  The key is difficult to discern as there seems to be only one shade of red in the table, while it specifies a light and a dark shade of red for increased ORs which are not and are statistically significant (p<0.05).

 Answer: Thank you for your feedback. In the submitted Figure, the colors result to be distinguished, but we understand that there may be some difficulties in reading them in the submitted paper. For clarity we changed the light-red color of Figure 1. We added a Supplementary Table (ST1) with ORs and CIs (see attachment).

In the last paragraph of the results- where risks are reported by time period, please add the confidence intervals for all stated ORs.

Answer: We added the CIs.

So that the choice of analytical approach is clear- i.e. why incidence rates were not calculable-, consider adding the statement on lines 210-212 to the methods.

 Answer: Thank you. We added this statement to the text. “Information on the total number of migrants living in the four provinces covered by the Cancer Registry were not available. We therefore could not calculate incidence rates and ratios directly comparing migrants and non-migrants, but rather used the proportions of cancer occurred in the two populations. In specific, we calculated the proportion of new cases of each cancer over total cancers among migrants and compared it with the same proportion among non-migrants.”

As different patterns are seen by different countries of origin, has there been- or are there anticipated to be- shifts in the origins of migrants to Sicily over the years?  If so, is this worthy of comment that the observed cancer patterns are reflective of the present situation but may change in the future?

Answer: Thank you for the stimulating questions. Migration to Southern Italy, including Sicily, is a public concern which calls for large efforts from Italian governments, and depends on several socioeconomic factors which may change in the time. Anyway, this is an acute problem rather than a chronic one, as most migrants to Sicily are quickly moving to other countries, not impacting cancer rates. We added this to the discussion section. “These data may be a useful source of information for understanding cancer epidemiology in migrants in Italy. Cancer control in this special population requires public attention, despite its future trends are unpredictable given the acute character of migration in Sicily, that is most migrants move quickly to other countries.”

We also added in the conclusions: “Cancer control and early detection in migrants may improve, while it is difficult to establish a follow-up on such special population which is quickly moving from Sicily to other countries.”

Reviewer 3 Report

Manuscript (ID: cancers-2324895) presents results of investigation of incidence cancer rates in the population of migrants in Sicily, Southern Italy. But, some issues in this manuscript require major revision:   

  • Line 12: In Background in Abstract, before the aim of this study, in one sentence, give an overview of the importance of this research. 

  • Lines 12-13: Starting from the title and objective of this manuscript, in the entire text of the manuscript, add the word `new' before the word `cases', and add the word `incidence' before the word `rate'. Rationale: An important strength of this study is the analysis of cancer incidence data, which should be stated in the paper as a whole (in text, Tables and Figure).  

  • Lines 21-22: For the precision, in this sentence it must be emphasized that the increased proportion of lung cancer refers to both sexes together (in the abstract the data is already given: PIR=1.20, 95% CI=1.07-1.33) and on females (PIR=1.32, 95% CI=1.11-1.56). 

  • Lines 25-28: For the sake of precision, in addition to the above, add `increase was also detected for lung cancer, in particular in women' in Conclusions. 

  • Lines 31-75: In the Introduction section, the most important information about the topic of the manuscript is presented, which is supported by citing relevant references. Also, the authors emphasized the importance of obtaining the results of such research, especially for the purpose of creating cancer prevention and control measures in this vulnerable population. Finally, the authors emphasized the need for current and similar research in order to reduce inequalities in cancer management in migrant and non-migrant populations. 

  • Lines 76-81: The aim of the manuscript is well defined. The suggestion is to move the sentence on Lines 78-81 to Conclusion at the end of this manuscript.  

  • Line 82: In section Methods add subsections, such as:  
    • `Study design`,
    • `Study setting`,
    • `Study population`,
    • `Data sources`,
    • `Statistical analysis` with defining the level of statistical significance for all applied tests,
    • `Ethical consideration`.  

  • Lines 205-209: It is pleasing to see that the Discussion section begins with a paragraph that should summarize the presented results. However, it is necessary to check and correct the given data. Namely, in section Results it is stated `decreased OR was suggested for liver, breast and prostate cancer', which is in contradiction with what was stated in first sentence in section Discussion, that is with `a lower proportion of breast and prostate cancers compared to non-migrants.`. To correct. Rationale: `decreased OR was suggested' and `a lower proportion' do not have the same meaning. It is necessary to highlight important statistically significant results, which is not shown in this sentence.   

  • Lines 210-218: Information on data quality should be described in the Methods section, where `Study population' and `Data sources' should be defined. Additionally, the information of the lack of population data on the number of migrants in Sicily must be discussed in Limitation of this study at the end of the Discussion section. 

  • Lines 231-232: Give a possible explanation for the presented differences in risk of liver cancer between migrant men and migrant women.  

  • Lines 235-239: Give a possible explanation for the presented differences in incidence of lung cancer between migrant men and migrant women. 

  • Lines 242-246: Give a possible explanation for the differences in breast cancer incidence patterns in migrant women originating from different regions.   

  • Line 319: Mandatory to add a new paragraph detailing the limitations of this study. Discuss other issues regarding study design, data quality, lack of data of some variables such as education level, place of residence, marital status, etc.   

Author Response

Comments and Suggestions for Authors

Manuscript (ID: cancers-2324895) presents results of investigation of incidence cancer rates in the population of migrants in Sicily, Southern Italy. But, some issues in this manuscript require major revision:   

  • Line 12: In Background in Abstract, before the aim of this study, in one sentence, give an overview of the importance of this research. 

Answer: Done.

  • Lines 12-13: Starting from the title and objective of this manuscript, in the entire text of the manuscript, add the word `new' before the word `cases', and add the word `incidence' before the word `rate'. Rationale: An important strength of this study is the analysis of cancer incidence data, which should be stated in the paper as a whole (in text, Tables and Figure).  

    Answer: Thanks for the suggestions. We accounted for yours and other reviewers’ comments, one regarding the use of the terms “proportionate morbidity ratio” rather than “incidence”, and edited the paper in consequence.

  • Lines 21-22: For the precision, in this sentence it must be emphasized that the increased proportion of lung cancer refers to both sexes together (in the abstract the data is already given: PIR=1.20, 95% CI=1.07-1.33) and on females (PIR=1.32, 95% CI=1.11-1.56). 

Answer: Done, thanks.

  • Lines 25-28: For the sake of precision, in addition to the above, add `increase was also detected for lung cancer, in particular in women' in Conclusions. 

Answer: Done. Our concern is the length of the abstract in this current version, to be agreed with the editors.

  • Lines 31-75: In the Introduction section, the most important information about the topic of the manuscript is presented, which is supported by citing relevant references. Also, the authors emphasized the importance of obtaining the results of such research, especially for the purpose of creating cancer prevention and control measures in this vulnerable population. Finally, the authors emphasized the need for current and similar research in order to reduce inequalities in cancer management in migrant and non-migrant populations. 

  • Lines 76-81: The aim of the manuscript is well defined. The suggestion is to move the sentence on Lines 78-81 to Conclusion at the end of this manuscript.  

Answer: We agree, thanks.

  • Line 82: In section Methods add subsections, such as:  
    • `Study design`, 
    • `Study setting`, 
    • `Study population`, 
    • `Data sources`, 
    • `Statistical analysis` with defining the level of statistical significance for all applied tests, 
    • `Ethical consideration`.  

Answer: We followed your suggestion.

  • Lines 205-209: It is pleasing to see that the Discussion section begins with a paragraph that should summarize the presented results. However, it is necessary to check and correct the given data. Namely, in section Results it is stated `decreased OR was suggested for liver, breast and prostate cancer', which is in contradiction with what was stated in first sentence in section Discussion, that is with `a lower proportion of breast and prostate cancers compared to non-migrants.`. To correct. Rationale: `decreased OR was suggested' and `a lower proportion' do not have the same meaning. It is necessary to highlight important statistically significant results, which is not shown in this sentence.   

Answer: Thank you for the careful reading. We distinguished the PIR analysis (now PMR) from the multivariate analysis, which are overall consistent but present small differences. Among the differences there is the fact that while in the PMR analysis no pattern of risk is present for liver cancer, the multivariate analysis shows a decrease in the risk. Moreover, this decreased risk is of borderline significance (OR=0.80, 0.64-1.01). These are the reasons why we chose not to mention this in the summary in the discussion. We used the word “suggested” because of the borderline statistical significance of some values, but we edited the sentence to be more precise as you recommend. We corrected the sentence in the discussion referring to “odds” instead of “proportions”, in relation to the results shown in Table 3.

  • Lines 210-218: Information on data quality should be described in the Methods section, where `Study population' and `Data sources' should be defined. Additionally, the information of the lack of population data on the number of migrants in Sicily must be discussed in Limitation of this study at the end of the Discussion section. 

Answer: We added the sentence in the methods. The registry used is included in Cancer Incidence in Five Continents, that includes only registries of high-quality. Also, cases were identified only from death certificates. We added these details in the methods and added high data quality in the strength of the study. Regarding the second request, the point you raised is already present as main limitation: “First, as already mentioned, the number of migrants was not available, preventing us from calculating the incidence of cancer in migrants and compare it with that in non-migrants”. We only added “in Sicily” to specify we refer to the lack of regional-level data.

  • Lines 231-232: Give a possible explanation for the presented differences in risk of liver cancer between migrant men and migrant women.  

Answer: Thank you for the stimulating comment, we added a sentence referring to the possibility of a different frequency in medical examinations and tests in the sexes due to sociocultural factors.

  • Lines 235-239: Give a possible explanation for the presented differences in incidence of lung cancer between migrant men and migrant women. 

Answer: We already mentioned the fact that “The difference between women and men is likely due to the low incidence of lung cancer among women in Southern Italy [18], which is explained by low tobacco consumption in past decades”. Anyway, we added a sentence to further explain this hypothesis.

  • Lines 242-246: Give a possible explanation for the differences in breast cancer incidence patterns in migrant women originating from different regions.   

Answer: We added an hypothesis, namely possible differences in cancer screening participation and oral contraceptives use. 

  • Line 319: Mandatory to add a new paragraph detailing the limitations of this study. Discuss other issues regarding study design, data quality, lack of data of some variables such as education level, place of residence, marital status, etc.   

Answer: We are not sure regarding this point, as the limitation paragraph is already present in the manuscript: “This study has some limitations. First, as already mentioned, the number of migrants in Sicily was not available, preventing us from calculating the incidence of cancer in migrants and compare it with that in non-migrants. In addition, no information was available on the duration of residence of the migrants: analyses by duration of stay in the host country help to clarify the role of factors affecting cancer risk [64]. Moreover, the lack of information on lifestyle, occupational and sociodemographic factors prevented us from adjusting and stratifying for important variables. Also, results by area of origin of the migrant population showed patterns of risk which were not fully consistent with previous literature. Last, small numbers have impaired the stratified analyses for some areas of origin”.

The study is designed as a particular case-control study, where migrant status is the exposure and other cancers are the controls. Data quality was addressed above. We added a mention to educational level as a potential confounder for completeness. We performed a preliminary analysis stratified by province, which however did not suggest any consistent pattern of results. 

Round 2

Reviewer 2 Report

Thank you for addressing my comments.

Reviewer 3 Report

Thank you for the opportunity to re-review manuscript ID: cancers-2324895. In the revised version of this manuscript, the authors insert the significant changes and satisfactorily addressed all of my comments. Overall, regarding my comments:   

  • In revised version, the Abstract provides sufficient background about cancer rates in the population of migrants in Sicily, Southern Italy.      
  • In revised version, the research design is appropriate and the methods adequately described.
  • The Discussion section is significantly clearer and more complete. The Discussion section now is appropriate and supported by the satisfactory explanations for results in this study.   

I thank the authors.